# Dietary Sources of Linoleic Acid (LA) Differ by Race/Ethnicity in Adults Participating in the National Health and Nutrition Examination Survey (NHANES) between 2017–2018

**DOI:** 10.3390/nu15122779

**Published:** 2023-06-16

**Authors:** Shabnam R. Momin, Mackenzie K. Senn, Ani Manichaikul, Chaojie Yang, Rasika Mathias, Mimi Phan, Stephen S. Rich, Susan Sergeant, Michael Seeds, Lindsay Reynolds, Floyd H. Chilton, Alexis C. Wood

**Affiliations:** 1USDA/ARS Children’s Nutrition Research Center, Baylor College of Medicine, Houston, TX 77030, USA; shabnam.momin@usda.gov (S.R.M.); mackenzie.senn@uth.tmc.edu (M.K.S.); mimiphan96@gmail.com (M.P.); 2Center for Public Health Genomics, University of Virginia, Charlottesville, VA 22908, USA; am3xa@virginia.edu (A.M.); cy6n@virginia.edu (C.Y.); ssr4n@virginia.edu (S.S.R.); 3Department of Medicine, Johns Hopkins University, Baltimore, MD 21205, USA; rmathias@jhmi.edu; 4Department of Internal Medicine/Section on Molecular Medicine, Wake Forest School of Medicine, Winston-Salem, NC 27101, USA; ssergean@wakehealth.edu; 5Department of Biochemistry, Wake Forest School of Medicine, Winston-Salem, NC 27101, USA; mseeds@wakehealth.edu; 6Department of Epidemiology and Prevention, Wake Forest School of Medicine, Winston-Salem, NC 27101, USA; lireynol@wakehealth.edu; 7Department of Nutritional Sciences, University of Arizona, Tucson, AZ 85721, USA; schilton@wakehealth.edu

**Keywords:** food groups, meat, fish, grains, fruits and vegetables, adequate intake, cardiovascular disease, polyunsaturated fatty acids

## Abstract

Linoleic acid (LA) is a primary *n*-6 polyunsaturated fatty acid (PUFA), which is of interest to nutritional professionals as it has been associated with health outcomes. However, as some LA-rich foods offer protection against chronic diseases such as CVD (e.g., fatty fish), while others increase risk (e.g., red meat), the individual foods contributing to LA intake may be an important factor to consider. Therefore, this analysis sought to examine whether there are racial/ethnic differences in the proportion of overall LA intake accounted for by individual food groups, via a cross-sectional analysis of 3815 adults participating in the National Health and Nutrition Examination Survey (NHANES; 2017–2018 cycle). Separate multivariable linear regressions models specified the proportion of overall LA intake attributable to each of the nine food groups (dairy, eggs, fat, fish, fruits and vegetables, grains, meat, nuts, and sweets) as the outcome, and race/ethnicity as the predictor, with age, gender, and socioeconomic status (SES) as covariates, in order to estimate whether there were mean differences by race/ethnicity in the proportion of overall LA intake attributable to each of these foods seperately. After a Bonferroni correction for multiple testing, eggs, grains, fruits and vegetables, meat, and fish each accounted for a different proportion of overall LA intake according to racial/ethnic grouping (all *p* < 0.006 after a Bonferroni correction). These findings indicate the food sources of LA in the diet differ by race/ethnicity, and warrant future investigations into whether this plays a role in health disparities.

## 1. Introduction

The 1990 National Nutrition Monitoring and Research Related Act coded the importance of research efforts that seek to define the food intake of the U.S. into law, and protected the right of the U.S. Department of Agriculture (USDA) and Health and Human Services (HHS) to carry out activities in the pursuit of this [1]. The importance of continuously monitoring, and updating information on, the nutrition intake of U.S. citizens continues to be recognized by the USDA. This endeavor was named one of the key research areas in the human nutrition program for the 2016–2021 research cycle [2], and is proposed to remain as one of five of priority areas for the upcoming five-year cycle [3].

One potential benefit of national-level efforts to monitor food intake is the potential to gain information into differences in the intake of subpopulations within the U.S., which could subsequently yield insights into any health disparities between these groups. Linoleic acid (LA; 8:2n − 6) is an essential *n*-6 polyunsaturated fatty acid (PUFA) required for normal growth and development [4,5], which also plays a role in the prevention of chronic diseases such as cardiovascular diseases (CVD) [6]. Although recommended daily intakes (RDAs) for LA do not exist [7], reference data suggest that adequate intakes (AIs) of LA for women and men, respectively, are 12 g/d and 17 g/d for ages 19–50 years, and 11 g/d and 14 g/d for ages 51–70 years [8]. The American Heart Association recommends that the Acceptable Macronutrient Distribution Range (AMDR) of LA is 5–10% of total energy [9]. Population-level reports on overall LA intake showed typical intakes of LA around 12.6 g/d by adult women and 16.0 g/d by adult men in 2007 [10,11], corresponding to 5.5% and 6.0% of total average energy intake, respectively [11]. 

One report suggested that NHWs have higher LA intakes than NHBs and MAs (although this was not tested for significance [12]), indicating the potential for race/ethnic differences in LA and the consumption of LA-containing foods [12]. However, we are not aware of studies that have explicitly tested for the presence of such differences; i.e., for differences in consumption of LA-containing foods by race/ethnicity and differences in the contributions of these to overall LA intake. Therefore, the goal of the current study was to estimate whether the proportion of overall LA intake attributable to each of nine food groups (dairy, eggs, fat, fish, fruits and vegetables, grains, meat, nuts, and sweets) differed by race/ethnicity, using nationally representative data from the NHANES 2017–2018 cycle. Although there are known race- and ethnicity-driven differences in the dietary sources of other unsaturated fatty acids such as eicosapentaenoic acid (EPA) and docosahexaenoic acid (DHA) [13,14,15,16], no study has examined this for LA as a micronutrient.

## 2. Materials and Methods

### 2.1. Participants

NHANES contains data from an ongoing set of interviews and exams designed to assess the health and nutritional status of the civilian noninstitutionalized populations of the United States [17,18]. Certain groups, such as those older than 60 years and Hispanics, are oversampled to ensure that the data reflect current population trends [17,18]. In the current study, we used data from one cycle of NHANES (2017–2018), which was the latest released dataset with dietary measurements, to estimate the LA intake among U.S. adult participants aged over 20 years.

There were 7641 individual records of dietary data in the 2017–2018 dataset for individuals, and we excluded 2899 participants under 20 years of age. We further excluded pregnant and lactating women (*n* = 113), as well as individuals with implausible dietary intake data (defined as ≥600≤ kilocalories [Kcal] per day; *n* = 814), yielding a final sample size of 3815.

This study was approved by the Institutional Review Board at Baylor College of Medicine and was declared as non-human subject research, so it was exempted from a full review (protocol number: H-49021).

### 2.2. Measures

#### 2.2.1. Dietary Data 

In NHANES, dietary intake is assessed using up to two 24-h dietary recall (24DR) interviews from the What We Eat in America (WWEIA) survey, administered by a trained interviewer using the United States Department of Agriculture’s (USDA) Automated Multiple Pass Method [19]. The first 24DR was conducted at a Mobile Examination Center using a standard set of measuring guides to help estimate the portion size. A second 24DR was conducted on a subset (~80%) of the study population by telephone interview on a non-consecutive day within three to ten days. Participants were given measuring cups, spoons, a ruler, and a food model booklet comprising measuring guides given during the first interview to use for reporting food amounts during the telephone interview. Due to the possibility of unrepresentative results when only using a single 24DR, we used data averaged using both 24DR where possible.

Food group LA intake was estimated for nine food groups, which were selected based on previous analyses, showing these would account for almost all LA intake in the U.S. population [12]. The nine food groups included were dairy, eggs, fat, fish, fruits and vegetables, grains, meat, nuts, and sweets.

To derive LA intake by food group, first, the LA composition from each individual food/beverage reported as consumed in the NHANES population was calculated using USDA’s Food and Nutrient Database for Dietary Studies 2017–2018 (FNDDS 2017–2018) [20]. All foods were then ascribed to one of our nine food groups based on the FNDDS categories—dairy, meat, fish and shellfish, nuts and legumes, fats, grains, sweets, eggs, and fruits and vegetables. LA intake per day (in grams) for each food group was calculated as the sum of LA (in grams) from all foods ascribed to that group, and then converted to kilocalories of LA energy by multiplying the gram amount by a factor of nine. 

To calculate AMDR, the absolute LA intake in grams was converted to kilocalories by multiplying the gram amount by a factor of nine, reflecting the nine kilocalories contained in each gram of dietary fat, and expressed as the percentage of total kilocalories. 

#### 2.2.2. Demographic Variables

Self-reported age, sex, race/ethnicity, education level, and family income was collected by NHANES personnel during the household interview. Sex was coded by NHANES personnel as either male or female, and referred to as ‘gender’ in the NHANES literature, a term we use for the current report. Family income was operationalized using the family poverty-to-income ratio, which reflected the annual family income relative to the federal poverty level.

### 2.3. Statistical Analyses

All analyses were performed using R studio version 3.5.2 [21]. Sample weights were used to account for the complex survey sample design, and analyses were conducted using the ‘survey’ package. 

#### 2.3.1. Descriptive Statistics 

For descriptive statistics, means were calculated using the ‘svymean’ command and frequencies were calculated using the ‘svytable’ command. *T*-tests were used to examine differences in these factors by race/ethnicity for continuous variables (transformed to a normal distribution via an inverse normal transformation where necessary), and chi-square (χ^2^) test for categorical variables. In this exploratory analysis, differences were considered statistically significant at *p* < 0.05. Descriptive statistics are presented in Table 1.

#### 2.3.2. Differences by Race/Ethnicity in the Contribution of Each Food Group to Overall LA Intake

Differences by race/ethnicity for the contribution of each food group to the overall LA intake were examined using multivariable linear regressions, with the proportion of LA attributable to each of the nine food sources as the outcomes in separate models, racial/ethnic group as the predictor, and age, gender, poverty-to-income ratio, and education level as covariates. First, a global test of differences by race/ethnicity was conducted in models specified according to the following equation:y ~ x + covariates(1)
where y is the proportion of LA calories accounted for by a given food group and x represents race/ethnicity. All models were adjusted for the complex survey design via the use of sample weights. Significance was set at a Bonferroni-corrected *p* < 0.006 (0.05/9), and the results are presented in Table 2. 

Then, where a global difference by race/ethnicity was significant, post hoc pairwise comparisons were requested and adjusted for multiple testing using the Tukey-Kramer procedure; a post hoc test is suitable where group sizes differ. These results give an indication of which specific race/ethnic groups showed differences (and which did not), as well as the directionality of any group differences. These results are presented in Table 3. 

## 3. Results

### 3.1. Descriptive Statistics

Demographic characteristics and information on LA intake stratified by race/ethnicity are presented in Table 1. There were noticeable differences by race/ethnicity for age, education level, and poverty-to-income ratio (all *p* < 0.05; Table 1), but not gender (*p* > 0.05; Table 1). The mean LA intake was 17.9 g/d and AMDR was 7.66%. NHBs and NHWs reported the highest overall LA intake of 18.9 g/d and 18.25 g/d, respectively, and significantly differed from MAs (17.11 g/d), OHs (16.05 g/d), and those reporting other/mixed races (16.53 g/d; *p* < 0.05; Table 1). NHBs also had the highest AMDR of 8.2% and NHW had the second highest of 7.8%, both of which significantly differed from the AMDR in MAs (7.32%), OHs (6.86%), and those reporting other/mixed races (7.23%; *p* < 0.05; Table 1).

### 3.2. Contribution of Food Groups to Overall LA Intake by Race and Ethnicity

All food groups showed significant differences by race/ethnicity in terms of their contribution to overall LA intake in the global population tests (all *p* < 0.001; Table 2). 

#### 3.2.1. Differences in the Contribution of Dairy to Overall LA Intake by Race/Ethnicity

No pairwise comparisons for the contribution of dairy to overall LA intake reached significance, although the higher contribution in NHW compared NHBs (5.49% ± 0.33 vs. 3.49% ± 0.34; Table 1) trended towards significance (β = 1.46, SE = 0.47, *p* = 0.02; Table 3, Figure 1).

#### 3.2.2. Differences in the Contribution of Eggs to Overall LA Intake by Race/Ethnicity

Eggs made a higher contribution to overall LA intake in MAs (7.11% ± 0.7; Table 1) compared to NHWs (4.1% ± 0.28; β = 3.09, SE = 0.84, *p* = 0.002; Table 3, Figure 1) and NHBs (3.5% ± 0.27; β = 3.72, SE = 0.85, *p* = 0.001; Table 1 and Table 3). Eggs also made a higher contribution to overall LA intake in the OH group compared to NHBs (6.47% ± 0.70 vs. 3.5% ± 0.27 [Table 1]; β = 2.58, SE = 0.76, *p* = 0.005; Table 3, Figure 1). 

#### 3.2.3. Differences in the Contribution of Fat to Overall LA Intake by Race/Ethnicity

Fat made a lower contribution to overall LA intake in those reporting mixed/other race (8.16% ± 0.64; Table 1) compared to that of NHBs (12.26% ± 0.69; β = 4.29, SE = 1.03, *p* < 0.001; Table 1 and Table 3, Figure 1) and NHWs (12.72% ± 0.58; β = 4.38, SE = 0.93, *p* < 0.001; Table 1 and Table 3, Figure 1).

#### 3.2.4. Differences in the Contribution of Fish to Overall LA Intake by Race/Ethnicity

Only NHWs and NHBs showed significant differences in the extent that fish contributed to overall LA intake, with fish contributing a significantly higher proportion of overall LA intake in NHBs (5.59% ± 0.51 vs. 2.91% ± 0.3 [Table 1]; β = 2.82, SE = 0.66, *p* < 0.001; Table 3, Figure 1).

#### 3.2.5. Differences in the Contribution of Fruits and Vegetables to Overall LA Intake by Race/Ethnicity

Fruits and vegetables contributed less to overall LA intake in MAs (3.55% ± 0.66; Table 1) compared to NHWs (12.97% ± 0.55 [Table 1]; β = 3.71, SE = 1.03, *p* = 0.003; Table 3, Figure 1), NHBs (13.8% ± 0.55 [Table 1]; β = 3.94, SE = 0.97, *p* ≤ 0.001; Table 3, Figure 1), and those reporting mixed/other race (13.77% ± 0.62 [Table 1]; β = 4.29, SE = 1.05, *p* ≤ 0.001; Table 3, Figure 1).

#### 3.2.6. Differences in the Contribution of Grains to Overall LA Intake by Race/Ethnicity

Grains contributed to a greater proportion of overall LA intake in MAs (34.66% ± 1.32; Table 1) and OHs (32.89% ± 1.32; Table 1) compared to that of NHWs (22.21% ± 0.71 [Table 1], β = 7.67, SE = 1.70, *p* < 0.001; Table 3, Figure 1), NHBs (23.39% ± 0.77 [Table 1], β = 10.12, SE = 1.70, *p* < 0.001; Table 3, Figure 1), and those reporting other/mixed race (25.70% ± 1.00 [Table 1], β = 7.74, SE = 1.84, *p* < 0.001; Table 3, Figure 1).

#### 3.2.7. Differences in the Contribution of Meat to Overall LA Intake by Race/Ethnicity

MAs reported a relatively low contribution of meat to LA intake (15.34% ± 0.96; Table 1),compared to NHWs (17.06% ± 0.56 [Table 1]; β = 4.39, SE = 1.26, *p* = 0.005; Table 3, Figure 1), NHBs (24.78% ± 0.9 [Table 1]; β = 10.08, SE = 1.46, *p* < 0.001; Table 3, Figure 1), and those reporting mixed/other race (20.41% ± 1.30 [Table 1]; β = 7.06, SE = 1.81, *p* < 0.001; Table 3, Figure 1). The contribution of meat to overall LA intake was also lower in OHs (15.34% ± 0.93; Table 1) than in NHBs (24.78% ± 0.9 [Table 1]; β = 8.98, SE = 1.41, *p* < 0.001; Table 3, Figure 1) and those reporting mixed/other race (20.41% ± 1.30 [Table 1]; β = 5.95, SE = 1.74, *p* = 0.006; Table 3, Figure 1). The contribution of meat to overall LA intake was also higher in NHB than in NHWs (24.78% ± 0.90 vs. 17.06% ± 0.56 [Table 1]; β = 5.69, SE = 1.15, *p* ≤ 0.001; Table 3, Figure 1).

#### 3.2.8. Differences in the Contribution of Nuts to Overall LA Intake by Race/Ethnicity

The only significant race/ethnic differences in the contribution of nuts to overall LA intake was a lower contribution in NHBs compared to those reporting mixed/other race (9.38% ± 0.61 vs. 12.54% ± 0.91 [Table 1]; β = 5.59, SE = 1.10, *p* ≤ 0.001; Table 3, Figure 1).

#### 3.2.9. Differences in the Contribution of Sweets to Overall LA Intake by Race/Ethnicity

As with the contribution of nuts to overall LA intake by race/ethnicity, only one significant difference in the contribution of sweets to overall LA intake by race/ethnicity was found, in which sweets contributed a greater proportion to overall LA intake in NHWs compared to those reporting mixed/other race (10.11% ± 0.45 vs. 7.1% ± 0.53 [Table 1]; β = 2.50, SE = 0.73, *p* = 0.005; Table 3, Figure 1).

## 4. Discussion

In line with the USDA’s mission to more accurately document the food and nutrient intake of the U.S. population [2,3], the current analyses sought to identify the different sources of LA in the U.S. diet and examine whether the mean proportion of LA for each of nine specific food groups differed by race/ethnicity. Using recent data from NHANES (2017–2018) in the first study, of which we are aware, to use a large nationally representative sample to document differences in how different food sources contribute to overall LA intake in the U.S. diet, we found that race/ethnic differences in the proportion of LA contributed by all food groups, with the exception of dairy, which only showed a trend towards differences.

The greatest number of race/ethnic differences in the extent to which individual food groups accounted for overall LA intake were identified between MAs and NHWs, with the former group receiving more of their LA intake from eggs and whole grains, and less of their LA intake from fruits and vegetables and meat. MAs (along with OHs) also showed a moderate number of differences compared to NHBs, with NHBs receiving a lower portion of LA intake from eggs and a higher proportion from meat and fruits/vegetables. NHBs and NHWs were most similar in how individual food groups contributed to overall LA intake; only two differences were identified between these two groups, with NHBs receiving a lower proportion of LA from fish and a higher proportion of LA from meat. Although data on racial/ethnic differences in the contribution of food groups to overall LA has not been previously published, our results are concordant with other studies, suggesting that the foods contributing to other micronutrients differ among these groups, one study observed a stronger contribution of meat to cholesterol intake in NHBs and of grains in MAs compared to other racial/ethnic groups [14].

The USDA has denoted the surveillance of racial/ethnic differences in dietary intake as a ‘high priority’ area [2,3] due to the potential for understanding health disparities in diet-related risks for chronic diseases. It is beyond the scope of the present investigation to examine whether race/ethnic differences in the foods that contribute to overall LA intake could account for some of the disparities seen in CVD risk. However, it is notable that, for example, higher intakes of fish (although this may be limited to fatty fish) [22,23], dairy [24,25,26,27], fruits/vegetables [28,29,30], whole grains [31,32], and nuts [33,34,35] have been observed to be correlated with a lower risk of incident type 2 diabetes and CVD, while the intake of refined grains [31,36] and red meat [37,38] are generally observed to increase risk. Future studies should therefore examine whether race/ethnic differences identified by the current analyses may provide important insights into the risks of type 2 diabetes and CVD. If so, this may further help with personalizing nutrition advice for chronic disease prevention to an individual; for example, Mas had the lowest intakes of fruits and vegetables, suggesting that this group may benefit from interventions that target the intake of these foods. Similarly, NHWs had a much lower intake of fish than NHB, a potentially important consideration in ethnically-sensitive nutrition support. 

The use of NHANES data facilitated enough power to identify quality differences by race/ethnicity that are potentially generalizable to the U.S. population. However, the limitations of NHANES include the use of self-reported dietary intake data collected using 24DR, which contain sources of random and systematic errors. The use of trained investigators using a multi-pass technique in standardized protocols to collect the 24DR in the NHANES protocol may minimize some errors, but these can never be fully eliminated. 

## 5. Conclusions

The study highlights ethnic and racial differences in the mean proportion of LA attributable to various food groups in the U.S. population. These data may serve as a starting point for future investigations that seek to better understand health disparities in diet-related chronic diseases, and may help tailor health education and intervention programs aimed at improving health to the cultural backgrounds of diverse sub-populations within the U.S.

## Figures and Tables

**Figure 1 nutrients-15-02779-f001:**
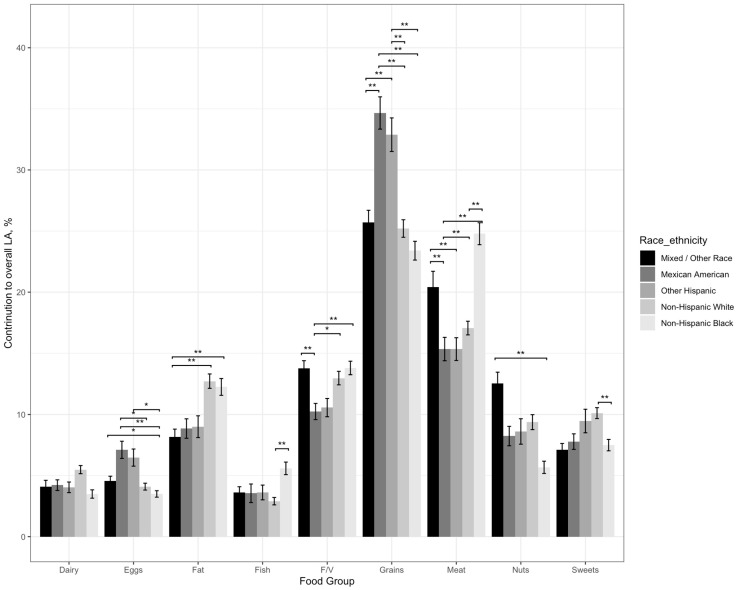
Percentage of overall LA intake attributable to each of the nine food groups, stratified by race/ethnicity, among adults in the NHANES 2017–2018 population. Note: Significance was set at a Bonferroni corrected *p* < 0.006. * = *p* < 0.006, ** *p* < 0.001 in models which controlled for age, gender, education level, and income level, and was adjusted for the complex survey design via sample weights.

**Table 1 nutrients-15-02779-t001:** Weighted means (±SE) or unweighted frequencies (with weighted percentages) for Demographic and Dietary Characteristics of adults in the NHANES population (2017–18).

	MA (*n* = 474)	OH (*n* = 341)	NHW (*n* = 1408)	NHB (*n* = 929)	Other (*n* = 663)
Demographics					
Age (years) ^a,b,e,f,g,h,i^	42.35 (0.89)	45.16 (1.13)	51.46 (0.63)	46.02 (0.68)	44.90 (0.87)
Gender, Female (*n*, %)	252 (52%)	179 (49%)	703 (51%)	473 (51%)	325 (50%)
PIR ^a,d,e,f,g,h,i^					
1.3 (%)	130 (29%)	107 (33%)	285 (13%)	246 (35%)	105 (23%)
>1.3–3.5 (%)	192 (47%)	104 (33%)	574 (32%)	336 (40%)	206 (37%)
>3.5 (%)	87 (24%)	81 (34%)	444 (55%)	193 (25%)	283 (40%)
Education ^a,b,c,d,e,f,h,i^					
9th grade (%)	119 (14%)	60 (11%)	25 (1%)	19 (2%)	24 (2%)
Less than HS (%)	85 (13%)	43 (9%)	123 (5%)	107 (10%)	34 (5%)
HS (%)	107 (34%)	68 (27%)	386 (26%)	235 (31%)	93 (22%)
Some college (%)	123 (30%)	102 (27%)	528 (31%)	372 (36%)	166 (30%)
College or above (%)	39 (10%)	66 (25%)	346 (38%)	194 (21%)	345 (41%)
Dietary intake					
LA (kcals/day) ^b,c,d,e,f,g^	154 (5.23)	144 (5.83)	164 (3.29)	170 (3.65)	149 (4.90)
LA intake (grams) ^b,c,d,e,f,g^	17.11 (0.58)	16.05 (0.65)	18.25 (0.37)	18.91 (0.41)	16.53 (0.54)
AMDR for LA (%) ^a,b,c,d,e,f,g,h^	7.32 (0.15)	6.86 (0.16)	7.77 (0.12)	8.19 (0.12)	7.23 (0.13)
Dairy (% of LA kcals) ^a,e,f,g^	4.22 (0.44)	4.04 (0.43)	5.49 (0.33)	3.49 (0.34)	4.09 (0.52)
Eggs (% of LA kcals) ^b,c,e,f,i,j^	7.11 (0.7)	6.47 (0.7)	4.1 (0.28)	3.5 (0.27)	4.57 (0.38)
Fat (% of LA kcals) ^b,c,d,e,f,g^	8.85 (0.8)	9 (0.89)	12.72 (0.58)	12.26 (0.69)	8.16 (0.64)
Fish (% of LA kcals) ^a,b,c,d^	3.55 (0.76)	3.62 (0.6)	2.91 (0.3)	5.59 (0.51)	3.63 (0.46)
Fruits/vegetables (% of LA kcals) ^b,c,d,e,f,i,j^	10.25 (0.66)	10.56 (0.75)	12.97 (0.55)	13.8 (0.55)	13.77 (0.62)
Grains (% of LA kcals) ^b,c,d,e,f,i,j^	34.66 (1.32)	32.89 (1.37)	25.21 (0.72)	23.39 (0.77)	25.7 (1.0)
Meat (% of LA kcals) ^a,b,c,d,g,i,j^	15.34 (0.96)	15.34 (0.93)	17.06 (0.56)	24.78 (0.9)	20.41 (1.3)
Nuts (% of LA kcals) ^a,b,c,d,g,i,j^	8.24 (0.79)	8.61 (1.04)	9.38 (0.61)	5.68 (0.5)	12.54 (0.91)
Sweets (% of LA kcals) ^a,e,g,j^	7.78 (0.64)	9.47 (0.96)	10.11 (0.45)	7.49 (0.47)	7.1 (0.53)

Abbreviations: AMDR: Acceptable Macronutrient Distribution Range; kcals: kilocalories; NHB: Non-Hispanic Black; NHW: Non-Hispanic White; MA: Mexican American; OH: Other Hispanics; Other: All other races and/or mixed race; PIR: Poverty-to-income ratio. Note: ^a^ = *p* < 0.05 NHB vs. NHW, ^b^ = *p* < 0.05 NHB vs. MA, ^c^ = *p* < 0.05 NHB vs. OH, ^d^ = *p* < 0.05 NHB vs. Others, ^e^ = *p* < 0.05 NHW vs. MA, ^f^ = *p* < 0.05 NHW vs. OH, ^g^ = *p* < 0.05 NHW vs. Others, ^h^ = *p* < 0.05 MA vs. OH, ^i^ = *p* < 0.05 MA vs. Others, ^j^ = *p* < 0.05 OH vs. Others.

**Table 2 nutrients-15-02779-t002:** Parameter estimates from ANCOVA models examining differences by race/ethnicity in the contribution of nine food groups to overall intake of linoleic acid in the adult NHANES 2017–18 population.

	F	df	*p*
Dairy	2258	4	2.5 × 10^−4^
Eggs	3983	4	3.2 × 10^−6^
Fat	11,944	4	2.8 × 10^−7^
Fish	2661	4	8.2 × 10^−4^
Fruits/vegetables	4201	4	0.001
Grains	39,904	4	1.2 × 10^−4^
Meat	27,867	4	4.0 × 10^−10^
Nuts	10,008	4	1.8 × 10^−5^
Sweets	5599	4	2.8 × 10^−5^

All models control for age, gender, education level, and income level. Abbreviations: ANCOVA: Analysis of Covariance; NHANES: National Health and Nutrition Examination Survey.

**Table 3 nutrients-15-02779-t003:** Post hoc comparisons for differences by race/ethnicity in the contribution of nine different food groups to overall LA intake among adults in the NHANES population (2017–2018).

	Dairy		Eggs		Fat		Fish		Fruits/Vegetables		Grains		Meat		Nuts		Sweets	
	ß (SE)	*p* *	ß (SE)	*p* *	ß (SE)	*p* *	ß (SE)	*p* *	ß (SE)	*p* *	ß (SE)	*p* *	ß (SE)	*p* *	ß (SE)	*p* *	ß (SE)	*p* *
MA vs.																		
OH	−0.23 (0.67)	0.10	−1.14 (1.05)	0.80	−0.28 (1.32)	0.10	−0.22 (1.10)	0.10	0.92 (1.10)	0.92	−1.33 (2.06)	0.97	1.11 (1.47)	0.94	0.13 (1.33)	1.00	1.04 (1.23)	0.91
NHW	0.60 (0.56)	0.82	−3.09 (0.84)	**0.002**	2.65 (1.11)	0.18	−1.33 (0.88)	0.54	3.71 (1.03)	**0.003**	−7.67 (1.70)	**<0.001**	4.39 (1.26)	**0.005**	−0.47 (1.15)	0.99	1.14 (0.87)	0.68
NHB	−0.88 (0.6084)	0.59	−3.72 (0.85)	**<0.001**	2.56 (1.17)	0.18	1.49 (1.06)	0.61	3.94 (0.97)	**<0.001**	−10.12 (1.70)	**<0.001**	10.08 (1.46)	**<0.001**	−2.56 (0.97)	0.06	−0.80 (0.88)	0.89
Other	−0.62 (0.77)	0.93	−2.33 (0.91)	0.07	−1.72 (1.16)	0.57	−0.63 (0.10)	0.97	4.26 (1.0487)	**<0.001**	−7.74 (1.84)	**<0.001**	7.06 (1.81)	**<0.001**	3.03 (1.25)	0.10	−1.36 (0.93)	0.58
OH vs.																		
NHW	0.83 (0.56)	0.56	−1.95 (0.78)	0.08	2.94 (1.19)	0.10	−1.11 (0.74)	0.55	2.79 (1.04)	0.06	−6.35 (1.66)	**0.001**	3.28 (1.20)	0.05	−0.60 (1.36)	0.99	0.11 (1.12)	0.10
NHB	−0.65 (0.62)	0.83	−2.58 (0.76)	**0.005**	2.85 (1.24	0.14	1.71 (0.89)	0.29	3.01 (1.01)	0.02	−8.80 (1.68)	**<0.001**	8.98 (1.41)	**<0.001**	−2.69 (1.25)	0.19	−1.85 (1.13)	0.46
Other	−0.39 (0.75)	0.98	−1.19 (0.84)	0.59	−1.44 (1.21)	0.76	−0.41 (0.83)	0.99	3.36 (1.06)	0.01	−6.41 (1.82)	**0.004**	5.95 (1.74)	**0.006**	2.90 (1.47)	0.27	−2.40 (1.16)	0.23
NHW vs.																		
NHB	−1.46 (0.47)	0.02	−0.63 (0.40)	0.48	−0.09 (0.10)	>0.99	2.82 (0.66)	**<0.001**	0.22 (0.88)	0.10	−2.45 (1.12)	0.18	5.69 (1.15)	**<0.001**	−2.08 (0.97)	0.19	−1.95 (0.70)	0.04
Other	−1.22 (0.62)	0.28	0.76 (0.50)	0.53	−4.38 (0.93)	**<0.001**	0.70 (0.57)	0.73	0.57 (0.89)	0.97	−0.06 (1.32)	1.00	2.67 (1.50)	0.38	3.50 (1.17)	0.02	−2.50 (0.73)	**0.005**
NHB vs.																		
Other	0.25 (0.70)	0.10	1.39 (0.50)	0.04	−4.29 (1.03)	**<0.001**	−2.12 (0.75)	0.04	0.35 (0.91)	0.10	2.39 (1.36)	0.39	−3.02 (1.70)	0.38	5.59 (1.10)	**<0.001**	−0.55 (0.77)	0.95

Notes: * *p* values after post hoc correction using the Tukey Kramer procedure. All associations control for age, gender, education level, and income level. Significant differences after a Bonferroni correction (*p* < 0.006) in bold. Abbreviations: NHB: Non-Hispanic Black; NHW: Non-Hispanic White; MA: Mexican American; OH: Other Hispanics; Other: All other races and/or mixed race.

## Data Availability

All data are open access and available for download at url:https://www.cdc.gov/nchs/nhanes/index.htm (accessed on 4 June 2023).

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
