# Peer review of "Dietary Sources of Linoleic Acid (LA) Differ by Race/Ethnicity in Adults Participating in the National Health and Nutrition Examination Survey (NHANES) between 2017–2018"

_nutrients, 2023, doi:10.3390/nu15122779_

Round 1
Reviewer 1 Report
è Planning of the study:
o Using multiplicity correction is well-received
o Using the survey weights in analyses is commandable
o No discussion around the significance of the interactions between race and the control variables in multivariable models.
o Instead of Table-2, model-based average of LA for each racial group would have been much more informative. Table-2 just shows the significance of race in the model. In fact, Table-3 is already doing that in a pairwise manner.
o The food groups were not specific enough, Linoleic Acid (LA) in different types of food within the same group. Maybe it could have been more significant to study only one food group in detail. For example the one that has the highest amount of LA.
è Objectives :
1) Contributions of 9 different food groups in overall LA intake among the U.S. adult population -> it was reached. The contributions were clearly explained in the conclusions of the paper.
2) Is overall LA intake attributable to different food sources according to race/ ethnicity using nationally representative data from the NHANES 2017-2018 99 cycle? => not really clearly formulated goal.
è Discussion: the data collected on the gender differences and income differences was not really discussed.
The results of the statistical analysis of this study were in fact contradicted by the actual data about the CVD; at least, there is no consistent trend to support the hypotheses. Here, interaction models may help.
Conclusion : in general no real outcome was formulated in the conclusion.
è Grammar :
· Some sentences are too long and often fail to convey one clear message.
· Commas missing in several places. Language editing is recommended (P2, L95-96, P3, l114, among many examples).
Author Response
Response to Reviewer 1 Comments
Point 1: Using multiplicity correction is well-received.
Response 1: Thank you. We have retained this approach in the revised submission.
Point 2: Using the survey weights in analyses is commandable.
Response 2: This approach has also been retained in the revised submission.
Point 3: No discussion around the significance of the interactions between race and the
control variables in multivariable models.
Response 3: We wished to examine whether the contribution of each of nine food groups
to overall linoleic acid (LA) acid differed by race/ethnicity, while controlling for overall caloric
intake. Therefore, we constructed nine variables, each representing the proportion of overall
LA calories accounted for by the consumption of one of the nine food groups. Our statistical
testing examined whether the mean values of our constructed variables, differed by
race/ethnicity:
Proportion of LA calories accounted for by a food group ~ race/ethnicity + covariates
This is a main effect analysis whereby race/ethnicity is the only predictor, and so it was not
possible to include an interaction.
Point 4: Instead of Table-2, model-based average of LA for each racial group would have
been much more informative. Table-2 just shows the significance of race in the model. In
fact, Table-3 is already doing that in a pairwise manner.
Response 4: The model-based average for proportion of LA attributable to each of nine
food groups, stratified by race/ethnicity, is available in Table 1. Our analysis goal was to
examine whether there were significant differences in these variables by race/ethnicity. We
conducted two analyses to examine this question: (1) a global test of differences by
race/ethnicity for each food group - these parameter estimates are presented in Table 2;
and (2) where the global test was significant, we conducted post-hoc tests to examine which
specific race/ethnicity comparisons differed – these parameter estimates are presented in
Table 3. For clarification: model-based means are presented in Table 1, while the parameter
estimates (point estimate, error term, and accompanying p-value) for each of the models,
for the first and second analysis steps, are presented in Table 2 and Table 3, respectively.
2
Point 5: The food groups were not specific enough, Linoleic Acid (LA) in different types of
food within the same group. Maybe it could have been more significant to study only one
food group in detail. For example the one that has the highest amount of LA.
Response 5: The food group with the highest amount of LA (i.e., which accounted for the
greatest proportion of overall LA intake), was grains. This food group is studied in detail.
As we declared our analyses ahead of time in our funding application (i.e., to include all
food groups) it would be unethical not to present all analyses, since this could be seen as
‘fishing’ and onlt presenting ‘selected’ results.
Objectives :
Point 6: 1) Contributions of 9 different food groups in overall LA intake among the U.S.
adult population -> it was reached. The contributions were clearly explained in the
conclusions of the paper.
Response 6: We appreciate these kind words.
Point 7: 2) Is overall LA intake attributable to different food sources according to race/
ethnicity using nationally representative data from the NHANES 2017-2018 99 cycle? =>
not really clearly formulated goal.
Response 7: We realize that this was not a clear articulation of our original objective. We
have rephrased this objective to state
“The goal of the current study were to estimate whether the proportion of overall LA
intake attributable to each of nine food groups differed by race / ethnicity, among the
U.S. adult population, using nationally representative data from the NHANES 2017-
2018 cycle”
Point 8: The data collected on the gender differences and income differences was not
really discussed.
Response 8: Thank you for noticing this. We had originally planned to include analyses
which tested for differences by gender, but to avoid over-analyzing the data, decided to
only include differences by race/ethnicity in the final analysis plan. We apologize for not
updating the background section thoroughly, and have removed all mention of potential
gender differences from the manuscript.
Point 9: The results of the statistical analysis of this study were in fact contradicted by the
actual data about the CVD; at least, there is no consistent trend to support the hypotheses.
Here, interaction models may help.
Response 9: Thank you for this helpful suggestion. It is not possible to include an
interaction term in the models as we have only one predictor. This has been clarified in the
3
statistical methods section of the revised manuscript. Please see our response to point
number 3, above, for further detail.
Point 10: Conclusion : in general no real outcome was formulated in the conclusion.
Response 10: We believe that the novel descriptive information on the dietary sources
that contribute to overall LA intake, and finding of statistically significant differences
between the race/ethnic groups, is an important outcome for the nutrition community.
Comments on the Quality of English Language
Grammar :
Comment: Some sentences are too long and often fail to convey one clear message.
Response: We have revised the text and shortened several sentences.
Comment: Commas missing in several places. Language editing is recommended (P2,
L95-96, P3, l114, among many examples).
Response: We have used a scientific editor to revise the language of the manuscript.

Reviewer 2 Report
The manuscript by Momin et al. analyzed how the dietary sources of Linoleic Acid (LA) differ by race/ethnicity in adults participating in the National Health and Nutrition Examination Survey (NHANES). The motivation for this study to investigate whether racial/ethnic groups get a greater or lesser proportion of their LA from each of the nine food groups was to examine its correlation to the known population disparities to CVD in the US.
Although the study aims to address a significant issue in dietary health, the major weaknesses of this study does not qualify for recommendation for publication in Nutrients.
The first weakness is that the result of this study does not provide strong evidence for testable hypotheses relating to the correlation between the race/ethnic differences in the foods that contribute to overall LA intake and the disparities seen in CVD prevalence, as the authors mentioned in the Discussion section. Therefore, the major purpose of the analysis performed in this study was not fulfilled by meaningful or directional result, especially that the manuscript highly emphasized on this point in the Introduction section.
The second weakness is that both the premise and the analytical design of this study are over simplified. It is very well established that the prevalence of CVD in certain racial/ethnic groups is related to multiple and complex factors, such as genetics, food intake, exercise and other lifestyle factors (apart from age, gender, socioeconomic factors etc.), among which genetics plays a very significant role, especially when it comes to the focus of race/ethnic differences in CVD risk. The authors did not address this key factor in either the introduction or the discussion part. This oversight might cause misleading understanding of the original hypothesis/premise of this study. Furthermore, the authors provided a decent overview of previous studies on how LA contributes to (lowering, in general) the risk of CVD in the introduction. However, the methods in this study to dissect the LA content in each of the nine food groups, thus correlating the difference in food group intake (with a focus on LA content) and CVD risk in certain sub-groups of population might be too indirect. Another important factor that the authors did not discuss in the study was that each of the nine food groups has their own complex composition of macronutrients and micronutrients that might also contribute to their potential link with CVD risk.
The third general weakness was the overall quality of the presentation of this manuscript. There are multiple grammatical errors in the text (e.g. Line 69-75; Line 89-94; Line 327) and low quality of image of figures (Figure 1). Therefore, the manuscript does not represent the good quality to meet the journal’s high standards.
Overall, although this study provided some insights in the ethnic and racial differences in the extent that various food groups can account for LA in take in the US population, the scope of this manuscript is overall too preliminary to be recommended for publication for a high impact journal such as Nutrients.
There are multiple grammatical errors in this manuscript (a few of these was mentioned above in the detailed comments).
Author Response
Response to Reviewer 2 Comments
Point 1: The manuscript by Momin et al. analyzed how the dietary sources of Linoleic Acid
(LA) differ by race/ethnicity in adults participating in the National Health and Nutrition
Examination Survey (NHANES). The motivation for this study to investigate whether
racial/ethnic groups get a greater or lesser proportion of their LA from each of the nine
food groups was to examine its correlation to the known population disparities to CVD in
the US.
Response 1: We would like to clarify that the goal of the manuscript is to provide
information on dietary sources of LA only:
“The goal of the current study were to estimate whether the proportion of overall LA
intake attributable to each of nine food groups differed by race / ethnicity, among the
U.S. adult population, using nationally representative data from the NHANES 2017-
2018 cycle”
The relationship to CVD provides context for why this information is important, but is not a
stated goal of the manuscript. The revised manuscript now emphasizes the importance of
the manuscript’s goal in the first paragraph, to provide context and clarity.
“The 1990 National Nutrition Monitoring and Research Related Act coded the
importance of research efforts that seek to define the food intake of the US into law, and
protected the right of the US Department of Agriculture (USDA) and Health and Human
Services (HHS) to carry out activities in the pursuit of this1
. The importance of
continuously monitoring and updating the nutrition intake of US citizens to providing
timely information about the contributions of food and nutrient consumption to health
continues to be recognized by the USDA, and this endeavor was named as one of the
key research areas in the human nutrition program for the 2016-2021 research cycle2
,
and is proposed to remain as one of five of priority areas for the upcoming five-year
cycle3
. ”
We also note the explicit comment in the original manuscript that correlations with CVD
risk were not a goal of the manuscript:
“It is beyond the scope of the present investigation to examine whether race/ethnic
differences in the foods that contribute to overall LA intake could account for some of
the disparities seen in CVD prevalence”
Point 2: Although the study aims to address a significant issue in dietary health, the major
weaknesses of this study does not qualify for recommendation for publication in Nutrients.
The first weakness is that the result of this study does not provide strong evidence for
testable hypotheses relating to the correlation between the race/ethnic differences in the
2
foods that contribute to overall LA intake and the disparities seen in CVD prevalence, as
the authors mentioned in the Discussion section. Therefore, the major purpose of the
analysis performed in this study was not fulfilled by meaningful or directional result,
especially that the manuscript highly emphasized on this point in the Introduction section.
Response 2: Our hypothesis is that there are racial and / or ethnic differences in the
relative contribution of each of several food groups to overall LA intake. We have used
large-nationally representative data to test this, and use frequentist tests (linear models)
with a Bonferroni correction to test this hypothesis. The linear models allow for positive and
negative betas (directions of effects) and in the discussion the direction of effect is
discussed e.g., [note: emphasis added to highlight the direction of results]:
“NHBs receiving a lower proportion of LA from fish and a higher proportion of LA from
meat compared to NHWs”
Point 3: The second weakness is that both the premise and the analytical design of this
study are over simplified. It is very well established that the prevalence of CVD in certain
racial/ethnic groups is related to multiple and complex factors, such as genetics, food
intake, exercise and other lifestyle factors (apart from age, gender, socioeconomic factors
etc.), among which genetics plays a very significant role, especially when it comes to the
focus of race/ethnic differences in CVD risk. The authors did not address this key factor in
either the introduction or the discussion part. This oversight might cause misleading
understanding of the original hypothesis/premise of this study. Furthermore, the authors
provided a decent overview of previous studies on how LA contributes to (lowering, in
general) the risk of CVD in the introduction. However, the methods in this study to dissect
the LA content in each of the nine food groups, thus correlating the difference in food
group intake (with a focus on LA content) and CVD risk in certain sub-groups of population
might be too indirect. Another important factor that the authors did not discuss in the study
was that each of the nine food groups has their own complex composition of
macronutrients and micronutrients that might also contribute to their potential link with CVD
risk.
Response 3: As above, the goal is not to correlate LA intake with CVD risk, as evidenced
by inclusion of individual-level data on (1) foods contributing to LA intake; and (2)
race/ethnicity; coupled with no inclusion of data on CVD risk. We also draw the reviewer’s
attention to the following comment:
“It is beyond the scope of the present investigation to examine whether race/ethnic
differences in the foods that contribute to overall LA intake could account for some of
the disparities seen in CVD prevalence”
3
Point 4: The third general weakness was the overall quality of the presentation of this
manuscript. There are multiple grammatical errors in the text (e.g. Line 69-75; Line 89-94;
Line 327) and low quality of image of figures (Figure 1). Therefore, the manuscript does
not represent the good quality to meet the journal’s high standards.
Response 4: This point is well taken and an independent scientific editor has reviewed the
manuscript for language and clarity.
Point 5: Overall, although this study provided some insights in the ethnic and racial
differences in the extent that various food groups can account for LA in take in the US
population, the scope of this manuscript is overall too preliminary to be recommended for
publication for a high impact journal such as Nutrients.
Response 5: If the goal of the manuscript was to link LA intake to CVD risk, we would
agree. However, the goal of the manuscript was stated as examining whether the foods
that contribute to LA intake differ by race/ethnicity. We believe frequentist tests using
large-scale nationally representative data provide generalizable scientific conclusions

Round 2
Reviewer 1 Report
Revisions address the comments and suggestions from this reviewer.
Reviewer 2 Report
The revised manuscript by Momin et al. sufficiently addressed the previous weakness in the flow of the text that could have misled the readers by overly emphasizing the correlation between LA intake and the risk of CVD, which is not the primary aim of this current study. The revision also provided more details in the robustness of the statistical analysis regarding the differences by race/ethnicity in the contribution of each food group to overall LA intake. The authors also addressed the other points in the previous review report. Overall, the clarification and revision significantly improved the quality of the manuscript and I would recommend the revised manuscript for publication.